# Pharmacokinetics of Biopharmaceuticals: Their Critical Role in Molecular Design

**DOI:** 10.3390/biomedicines11051456

**Published:** 2023-05-16

**Authors:** Takuo Ogihara, Kenta Mizoi, Akiko Ishii-Watabe

**Affiliations:** 1Laboratory of Clinical Pharmacokinetics, Graduate School of Pharmaceutical Sciences, Takasaki University of Health and Welfare, 60 Nakaorui-machi, Takasaki 370-0033, Japan; 2Faculty of Pharmacy, Takasaki University of Health and Welfare, 60 Nakaorui-machi, Takasaki 370-0033, Japan; mizoi-k@takasaki-u.ac.jp; 3Division of Biological Chemistry and Biologocals, National Institute of Health Sciences, 3-25-26 Tonomachi, Kawasaki-ku, Kawasaki 210-9501, Japan; watabe@nihs.go.jp

**Keywords:** biopharmaceuticals, pharmacokinetics, kinetic innovations, polyethylene glycol modification, antibody-related drugs, targeting

## Abstract

Biopharmaceuticals have developed rapidly in recent years due to the remarkable progress in gene recombination and cell culture technologies. Since the basic structure of biopharmaceuticals can be designed and modified, it is possible to control the duration of action and target specific tissues and cells by kinetic modification. Amino acid sequence modifications, albumin fusion proteins, polyethylene glycol (PEG) modifications, and fatty acid modifications have been utilized to modify the duration of action control and targeting. This review first describes the position of biopharmaceuticals, and then the kinetics (absorption, distribution, metabolism, elimination, and pharmacokinetics) of classical biopharmaceuticals and methods of drug quantification. The kinetic innovations of biopharmaceuticals are outlined, including insulin analog, antibody-related drugs (monoclonal antibodies, Fab analogs, Fc analogs, Fab-PEG conjugated proteins, antibody-drug conjugates, etc.), blood coagulation factors, interferons, and other related drugs. We hope that this review will be of use to many researchers interested in pharmaceuticals derived from biological components, and that it aids in their knowledge of the latest developments in this field.

## 1. Introduction

Biopharmaceuticals are drugs created by applying biotechnologies such as genetic recombinant and cell culture technology. Biopharmaceuticals have grown rapidly in recent years because of remarkable technological developments, making it possible to produce large amounts of protein present in trace amounts in living organisms or non-natural proteins with high purity. The authors calculated, based on a previous paper [1], that of the 143 drugs with global sales of more than USD 1 billion in 2019, sales of biopharmaceuticals accounted for more than 47% of the total.

The indications for the use of biopharmaceuticals are diverse, including cancer, rheumatism, and diabetes. They are also replacing traditional biopharmaceuticals such as blood coagulation factors (Figure 1) [2,3].

## 2. Kinetics of Pharmaceuticals Derived from Biological Components

Classic biopharmaceuticals are often derived from blood (plasma fractionated products). Blood coagulation factor VIII and IX, which are indicated for hemophilia, have traditionally been derived from fractionated human plasma. Ulinastatin and other drugs derived from human urine are also commercially available. This chapter describes the in vivo kinetics of common biopharmaceuticals, with an emphasis on classical biopharmaceuticals.

### 2.1. Absorption

Drugs derived from proteins and other biological components are rarely absorbed from the gastrointestinal tract in unchanged form due to their large molecular weight. With few exceptions, they are commercialized as injectable drugs because they are rapidly degraded by digestive enzymes in the gastrointestinal tract. They can be administered intravenously, intramuscularly, or subcutaneously. Drugs with a molecular weight of 5000 or more administered subcutaneously or intramuscularly tend to be absorbed via the lymphatic system, especially in drugs with a molecular weight of 20,000 or more, such as hematopoietic factor preparations [4], erythropoietin [5], and granulocyte colony-stimulating factor [6]. However, because of the extremely slow lymphatic flow rate, it takes time for drugs to reach the systemic circulation. For example, when antibody drugs are administered subcutaneously, the time to reach maximum blood concentration, T_max_, is 1 to 8 days from administration. Therefore, subcutaneous administration can maintain blood concentrations for a long period of time, but the pharmacological effect is not immediate. Subcutaneously administered biologic products may be degraded at the site of administration, resulting in a bioavailability of approximately 40% to 100%.

Drugs such as oral vaccines are absorbed from Peyer’s patches [7] (lymphoid immune organs located in the small intestine) in extremely small quantities. The body uses antigens absorbed from these organs to regulate immune responses such as antibody production.

### 2.2. Distribution

In general, drugs derived from high-molecular-weight biomolecules have difficulty permeating biological membranes and have low tissue distribution. Therefore, when administered intravenously, they tend to remain in the blood, and the volume of distribution is often approximately 60 mL/kg (body weight), equal to the plasma volume shown in the individual sections following Section 4. However, in organs with windowed (discontinuous) vessels such as the liver, macromolecules with molecular weights of up to 100,000 can be transferred relatively easily [8]. Therefore, pharmaceuticals derived from biological components may show heterogeneous distribution in the body.

### 2.3. Metabolism

High-molecular-weight pharmaceuticals such as proteins are translocated into cells by pinocytosis, phagocytosis by macrophages, monocytes, neutrophils, and receptor-mediated endocytosis, which are observed in many cells. If no specific intracellular transport pathway is involved, the drug is transferred to lysosomes, where it is metabolized by proteolytic enzymes. The liver and kidney also contain proteolytic enzymes with extracellular catalytic sites, such as aminopeptidase and γ-glutamyltranspeptidase. In addition, there are several soluble proteolytic enzymes in the blood that are responsible for the metabolism of high-molecular-weight drugs. Amino acids degraded by lysosomes are reused in vivo. In contrast with many low-molecular-weight drugs, which are metabolized by enzymes such as CYPs in the liver and gastrointestinal tract, many tissues are involved in the metabolism of high-molecular-weight drugs.

### 2.4. Elimination

Many pharmaceuticals derived from biocomponents are excreted in the urine after being metabolized into lower molecular weight compounds. Most of the examples given from Section 4 onwards are not excreted in the urine in their unchanged form. As with low-molecular-weight drugs, the ease of glomerular filtration of macromolecular drugs is related to their molecular size and charge, and the rate of filtration decreases markedly when the molecular weight exceeds 30,000. In addition, positively charged drugs are more easily filtered [9,10]. Relatively low-molecular-weight proteins filtered from the glomerulus are reabsorbed by endocytosis in the proximal tubules and broken down into amino acids in the tubular cells. Therefore, the unchanged form is rarely excreted in the urine, and even if it is, only a small per cent of the dose is excreted. However, the urinary excretion rate of preparations purified from human urine is high: 24% of the dose of ulinastatin (Table 1) is excreted in the urine [9].

The half-lives of plasma-derived drugs in the blood are 21–48 days for human immunoglobulins and 15–20 days for albumin. Factor VIII has the shortest half-life at 10–20 h. In contrast, the half-lives of urine-derived drugs are relatively short: 40 min for ulinastatin [9] and 17 to 33 min for urokinase [11,12] (Table 1).

Several antibody drugs and receptor analogs described below are eliminated by binding specifically to the target molecule, in addition to the elimination pathway of intracellular uptake by nonspecific cytokinesis. In this case, at low doses, the drug is rapidly eliminated from the blood by binding to the target molecule. At high doses, saturation occurs in binding to and uptake by the target molecule, and drugs that fail to bind to the target molecule are lost through nonspecific and slower elimination pathways, prolonging the half-life and resulting in nonlinear hemodynamics. This phenomenon has been observed with antibody drugs such as tocilizumab. The half-life of tocilizumab is 17 h at a dose of 0.15 mg/kg, but rises with an increase in dose, reaching 33 h at a dose of 0.5 mg/kg, 49 h at 1.0 mg/kg, and 74 h at 2.0 mg/kg (Figure 2, Table 2, [13]).

## 3. Drug Quantification and Evaluation of Blood Concentration

For the quantification of biological agents, methods based on immunoactivity assays and biological activity are selected. In animal studies, the radiolabeled method using iodine-125 (^125^I) is also used [14]. In either method, if the structure of the drug product is exactly the same as that of the biological components, it is difficult to completely separate and quantify them. Therefore, it is necessary to “subtract the biocomponents as background”. In such cases, precise quantitative standardization (validation) tests are necessary to set detection and quantification limits, and individual differences, gender differences, and circadian rhythms (physiological diurnal variation) of the biological components must also be considered.

ELISA is the most commonly used immunological method for measuring blood levels, followed by RIA. However, in this case, immune activity does not always reflect only unchanged metabolites, and it is possible that some metabolites retain immune activity but lose biological activity, or vice versa. It cannot be ruled out that some metabolites may enhance immune activity or biological activity. Therefore, it is desirable to investigate the correlation between immune activity, biological (pharmacological) activity, and unchanged drug amount (concentration) at least in an in vitro test system. On the other hand, if sensitivity and accuracy are sufficient, biological (pharmacological) activity can be taken as an indicator of drug concentration. In such a case, the activity does not necessarily reflect the concentration of the unchanged drug itself, but if the biological activity, including metabolites, is considered important, it is worth considering this activity in regard to the apparent unchanged drug amount (Figure 3).

## 4. Kinetic Innovation of Biopharmaceuticals

Since the basic structure of biopharmaceuticals can be designed and modified, it is possible to control the duration of action and target specific tissues and cells by modifying the kinetics. Examples of amino acid sequence modifications that extend or shorten the half-life of biopharmaceuticals include insulin preparations. There are also albumin fusion proteins, polyethylene glycol (PEG) modifiers, and fatty acid modifiers that bind to albumin in the blood, such as alptrepenonacog alpha, peginterferon alpha-1a, and GLP-1 analogs, which bind blood coagulation factor IX and albumin, respectively. GLP-1 analogs such as liraglutide are known. Collectively, these fusion proteins and their modifications have improved stability against proteolytic enzymes and prolonged half-life by inhibiting glomerular filtration. They are also expected to offer advantages in formulation with improved water solubility and safety-reduced antigenicity.

Monoclonal antibodies are single antibodies that bind specifically to disease-related molecules, taking advantage of the characteristic targeting and long half-life of antibodies. In recent years, the development of monoclonal antibodies as molecular-targeted drugs has progressed rapidly, and the number of drugs on the biopharmaceutical market is the largest in recent years. Antibody–drug conjugates are complexes of specific drugs attached to these monoclonal antibodies. Fc fusion proteins are proteins in which the Fc domain of an antibody is fused to a functional protein such as a receptor, peptide, enzyme, etc. Etanercept, which binds the Fc domain to the TNF receptor, Romiplostim, which binds the human thrombopoietin receptor binding sequence, and Phosphatase, which binds the Fc domain to the phosphatase, have been developed. Asphotase alpha bound to phosphatase is also known. The half-life and dosing intervals of these antibodies have been extended by binding to the Fc domain. Antibody Fab and receptor analogues exert their pharmacological effects by binding to specific target molecules and are intended for kinetic targeting.

Several drugs combine these approaches to simultaneously control the duration of action and targeting. Certolizumab pegol consists of the Fab domain of a humanized anti-TNFα monoclonal antibody bound to two molecules of PEG, and the Fab domain binds specifically to TNFα. The PEGylation of the Fab domain is also expected to enhance the long-lasting effect (Table 3).

### 4.1. Insulin

In 1923, an insulin preparation extracted from porcine pancreas was introduced to the market, making possible the pharmacological treatment of diabetes mellitus. Furthermore, 1958 saw the determination of the structure of insulin, and 1982 saw the development of a recombinant human insulin preparation, which greatly improved the convenience of insulin preparations. Insulin preparations need to supplement the basal and additional secretion of insulin in order to achieve the desired blood concentration trend. For this reason, super-fast-acting, rapid-acting, sustained-acting, and mixed (intermediate) types of insulin preparations have been developed.

Human insulin has a short duration of action and must be injected subcutaneously before each meal. However, because it can be sustained by forming crystals with protamine, once-daily protamine preparations or mixtures of these preparations and insulin are now on the market. These human insulin preparations do not take effect until approximately 30 min after administration. Therefore, patients need to administer insulin 30 min before a meal, but there is a risk of hypoglycemia if patients do not eat 30 min after receiving insulin. To avoid this, recombinant, ultra-fast-acting insulin analogs have been developed by modifying some of the amino acids in insulin. The insulin monomer consists of an A chain with 21 amino acids and a B chain with 30 amino acids. In the formulation, it forms a hexamer, which dissociates into a monomer in the subcutaneous tissue and enters the blood vessels where it exerts its effects. Insulin lispro [15] has a structure in which proline and lysine at positions 28 and 29 of the human insulin B chain are replaced. Although it exists as a hexamer in the formulation, it dissociates quickly into a monomer after subcutaneous injection, resulting in rapid transfer from the subcutaneous region into the bloodstream. Insulin glulisine [16,17] has asparagine at position 3 of the B chain replaced with lysine, and lysine at position 29 of the B chain replaced with glutamic acid. These substitutions make insulin glulisine more stable as a monomer and inhibit the aggregation of monomeric to dimeric and the dimeric to hexamic forms of insulin glulisine. In insulin aspart [18,19], proline at position 28 of the B chain is replaced with aspartic acid. In the formulation, the hexamer is weakly bound by zinc ion or phenol, but after subcutaneous injection, the hexamer is rapidly dissociated into dimers and monomers by dilution with body fluid and rapidly transferred into the bloodstream. Since these recombinant forms can also be sustained by forming crystals with protamine, a “mixed (intermediate) type” of the recombinant alone and a protamine preparation has been developed (Figure 4 and Figure 5, Table 4).

Sustained insulin analogs have also been developed to supplement basal secretion. These sustained insulin analogs are slowly and continuously transferred into the blood because they tend to remain stable hexamers in subcutaneous tissues and gradually dissociate into monomers. Insulin glargine [20] has an asparagine at position 21 of the A chain converted to glycine and two arginines added to the C terminus of the B chain. Due to these amino acid substitutions, the preparation is an acidic (pH 4) colorless solution, but its solubility is low in the neutral range. Subcutaneously administered, it is neutralized to form a fine insoluble precipitate from which insulin glargine is gradually released, resulting in a sustained action for approximately 24 h. Insulin detemir [21,22,23] is an insulin analog consisting of a C14 fatty acid side chain attached to lysine at position 29 of the B chain of human insulin. It binds to albumin at the injection site and in the blood via the fatty acid side chain, contributing to sustained action. Insulin degludec [24] lacks threonine at position 30 of the B chain and is composed of hexadecanedioic acid attached to a lysine residue at position 29 of the B chain using glutamic acid as a spacer (linker). In the formulation, it exists as a soluble di-hexamer, but after administration, it aggregates in the subcutaneous tissue to form a soluble and stable multi-hexamer, which temporarily remains in the subcutaneous tissue at the injection site. The monomer gradually dissociates from the multi-hexamer and is continuously absorbed into the blood from the site of administration. As with detemir, the drug binds to albumin at the injection site and in the blood via fatty acid side chains, resulting in sustained action.

### 4.2. Antibody-Related Drugs

Antibody-related drugs are structurally classified into IgG-type drugs that combine targeting to specific tissues and in vivo stability, Fab analogs and ScFv (single-chain antibodies) that are designed for kinetic targeting to specific tissues, and Fc analogs that are designed for stability in the body. Fab-PEG complex proteins with improved in vivo stability of Fab analogs and Fc fusion proteins with targeting and pharmacological functions added to Fc analogs are also subdivided. Antibody–drug conjugates, in which a drug is conjugated to a monoclonal antibody, are also included in antibody-related drugs (Figure 6).

Monoclonal antibodies are highly stable in vivo and have a half-life of over 24 h in the blood. This is in contrast to many small-molecule drugs, which need to be administered more than once a day. The volume of distribution is approximately 50–70 mL/kg (3.0–4.2 L assuming a body weight of 60 kg), which is equivalent to the volume of human plasma. Therefore, the drug is distributed almost entirely in plasma after administration, and its tissue transfer is considered to be relatively low (Table 5).

An antibody-drug conjugate consists of an antibody portion that targets an antigen protein (e.g., HER2) expressed in specific tissues such as cancer, a portion that has pharmacological effects such as anticancer activity, and a portion called a linker that bridges the two. Pharmacokinetic characteristics are similar to those of antibody drugs, with a long half-life in blood [54] (Table 6).

### 4.3. Blood Coagulation Factors

The possibility that products made from conventional biological materials may contain unknown pathogens cannot be denied, such as cases of human immunodeficiency virus and hepatitis C virus infections caused by blood products (plasma-fractionated products). Therefore, the switch from existing plasma-fractionated products to biopharmaceuticals is rapidly progressing. The blood coagulation factor VIII indicated for hemophilia has traditionally been derived from human plasma fractions, but recombinant versions of the octocog alpha and beta and the lulioctocog alpha have been developed. Although these recombinant forms have improved safety, they have not improved pharmacokinetic characteristics such as prolonged half-life. On the other hand, PEG-modified and Fc fusion proteins have extended half-lives and can be administered 1–2 times per week. The volume of distribution of these recombinant proteins is approximately 35–60 mL/kg, and they are distributed almost completely in plasma after administration (Table 7).

Recombinant blood factor IX has also been developed. Nonacog alpha and gamma do not show a significant increase in half-life compared to conventional plasma-fractionated preparations. On the other hand, PEG-modified, Fc-fused, and albumin-fused nonacog are longer-living and are administered once a week. The volume of distribution of nonacog beta-pegol is similar to that of plasma, while that of eftrenonacog alpha and albutrepenonacog alpha exceeds plasma volume, suggesting that nonacog beta-pegol is also translocated into the intercellular space and intracellular space.

### 4.4. Interferon

Interferon alpha (natural type) is a mixture of approximately 20 interferons produced by the mass culture of human lymphoblasts. Interferon alpha-2b is a recombinant form but has no pharmacokinetic modification compared to the natural form. Interferon alpha-2a has a different amino acid structure from alpha-2b. The PEGylated form of interferon alpha-2a is peginterferon alpha-2a, which has an extremely long half-life in blood and is administered once a week [70,71] (Table 8).

### 4.5. Other Examples

Etanercept, an inhibitor of human tumor necrosis factor TNF-alpha, is an Fc fusion protein in which the extracellular domain of the TNF receptor is bound to the Fc region of human IgG [72]. Similarly, abatacept, a drug for rheumatoid arthritis [73], and aflibercept beta, a drug for age-related macular degeneration, are Fc fusion proteins consisting of the receptor for the human vascular endothelial growth factor (VEGF) attached to the Fc region of an IgG antibody, respectively. These drugs have a long half-life in blood, contributing to a longer dosing interval.

Asphotase alpha, an enzyme preparation, is an Fc fusion protein that combines the catalytic domain of alkaline phosphatase with the Fc domain of human IgG [74]. Elapeguadermase is a PEG-modified form [75]. Dornase alpha is administered by inhalation using a nebulizer [76].

Erythropoietin, a hematopoietic factor used to treat renal anemia in dialysis patients, is a glycoprotein secreted by the kidneys in vivo. Since 2001, the appearance of darbepoetin alpha [77,78], in which five amino acid residues of erythropoietin have been modified to increase the number of glycosylation sites, and epoetin beta pegol, a PEG-modified form, has significantly extended the half-life of the drug [79]. The introduction of these new drugs has greatly extended their half-life. This has led to a significant reduction in the frequency of erythropoietin administration from two to three times a week to once every one to four weeks.

Filgrastim, a granulocyte colony-stimulating factor, is used in the treatment of neutropenia, and the PEG-modified form of filgrastim, pegfilgrastim has been used to extend its half-life in blood [80].

Human glucagon-like peptide-1 (GLP-1) is a gastrointestinal hormone that lowers blood glucose levels without hypoglycemia, especially in patients with type 2 diabetes. However, GLP-1 is rapidly degraded in vivo and has a short duration of action, making it unsuitable as a therapeutic agent. Liraglutide [81] is a fatty acid (hexadecanoic acid) modification of GLP-1. Semaglutide [82] is a modification of GLP-1 with some amino acids modified and a fatty acid (octadecanedioic acid) added. Dulaglutide [83] is an Fc fusion protein consisting of a modification of some amino acids in GLP-2 and the addition of Fc. These drugs are effective when administered once a week.

Biopharmaceuticals for human follicle-stimulating hormone and human chorionic gonadotropin are known, but they do not have improved kinetic characteristics [84,85,86]. Somatropin, a human growth hormone, must be administered once daily, but a formulation that fuses the substructures of both somatropin and human chorionic gonadotropin is commercially available as an analog of somatropin, somatrogon, and its administration frequency has been reduced to once a week with an extended half-life [87,88,89].

Although not included in biopharmaceuticals, proteins with a relatively small number of amino acid residues can be synthesized organically. For example, teriparatide, an osteoporosis drug, is a polypeptide consisting of 34 amino acids, and its acetate salt has been developed as a chemically synthesized product, while a recombinant form has been developed. The number of drugs in this category has grown in recent years as peptide synthesis technology has evolved dramatically with advances in combinatorics and automated synthesis, and they are sometimes called mid-molecular weight drugs because their molecular weight is approximately 1000 to 5000.

Somatostatin is a cyclic peptide consisting of 14 amino acids that inhibits the release of a growth hormone with a half-life of only a few minutes. Derivatives of somatostatin that shorten the amino acid chain and convert some amino acids from the natural ℓ-form to the d-form have prolonged the half-life in blood by escaping metabolism by degrading enzymes while maintaining the bioactivity of somatostatin. It is used in the treatment of gastrointestinal tumors and acromegaly (Figure 7).

## 5. Conclusions

This review summarizes the state-of-the-art biopharmaceuticals at this stage in terms of pharmacokinetic innovations. Biopharmaceuticals were initially developed with the main features of safety and mass production compared to conventional protein drugs. In recent years, there has been a qualitative shift toward biopharmaceuticals with improved in vivo pharmacokinetics. Biopharmaceuticals are a frontier field with the potential for the creation of drugs based on completely new concepts, such as mRNA vaccines. We look forward to further development of biopharmaceuticals in the future.

## Figures and Tables

**Figure 1 biomedicines-11-01456-f001:**
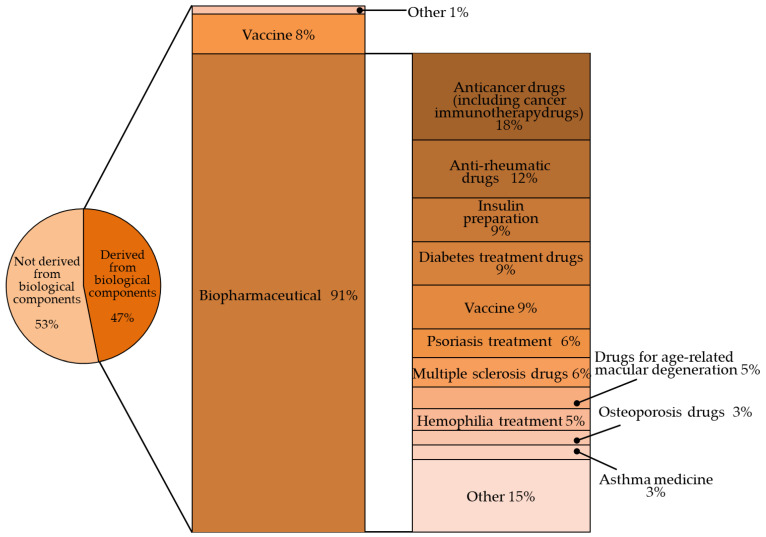
Percentage of biologic products among the world’s top-selling pharmaceutical products (143 products with sales of $1 billion or more in 2019) [1].

**Figure 2 biomedicines-11-01456-f002:**
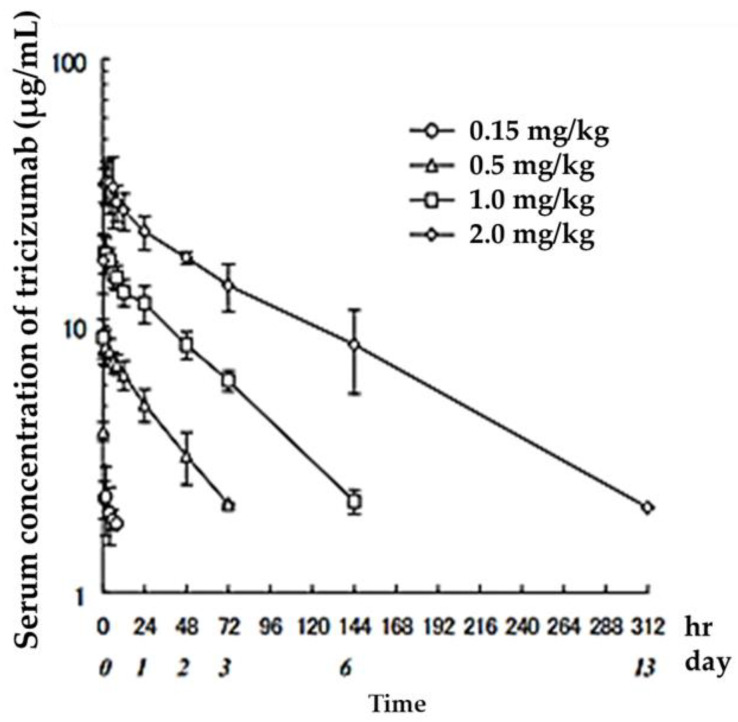
Serum concentrations of Tocilizumab in healthy adults after a single administration quoted from reference [13].

**Figure 3 biomedicines-11-01456-f003:**
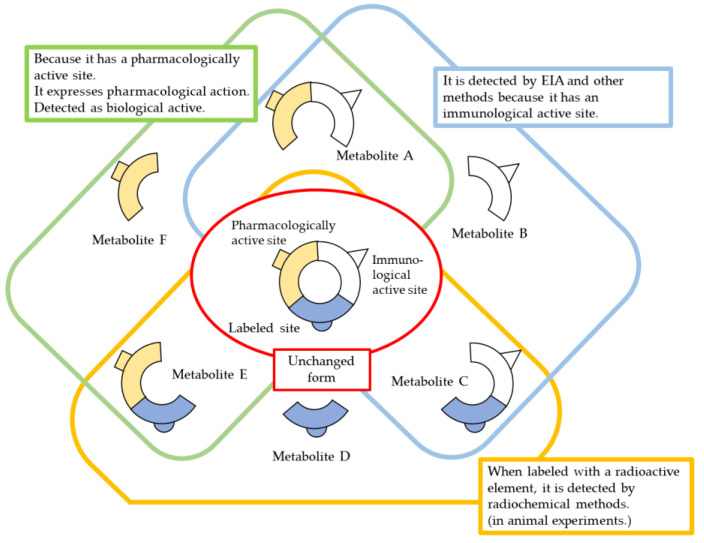
Methods and considerations for quantification of biopharmaceuticals.

**Figure 4 biomedicines-11-01456-f004:**
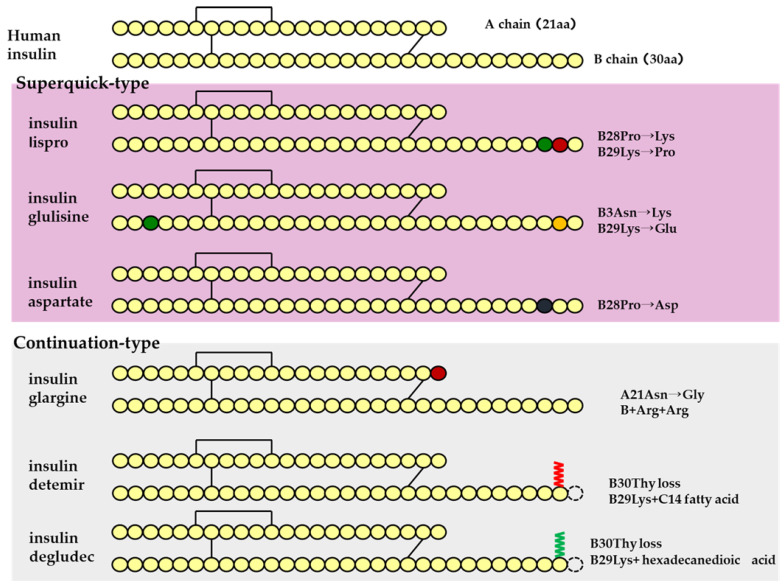
Structural schematic diagram of insulin preparation.

**Figure 5 biomedicines-11-01456-f005:**
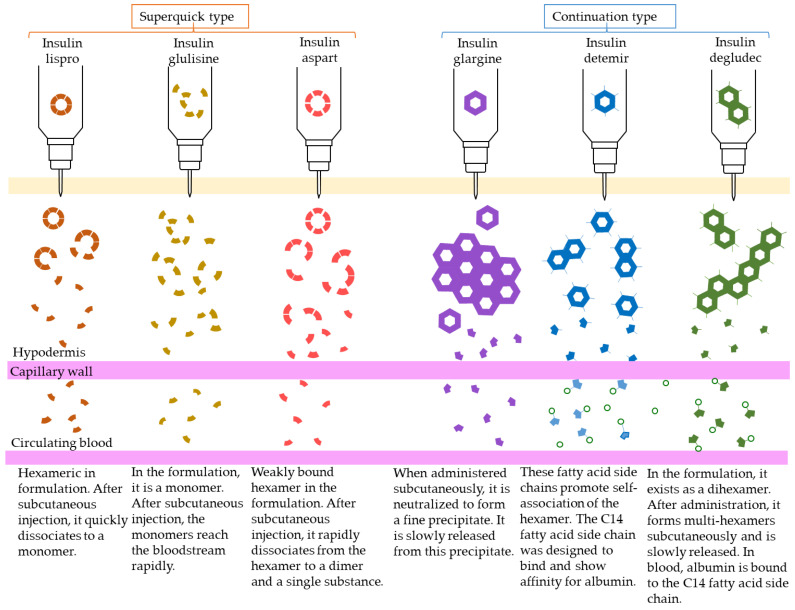
Mechanism of action of insulin analogs.

**Figure 6 biomedicines-11-01456-f006:**
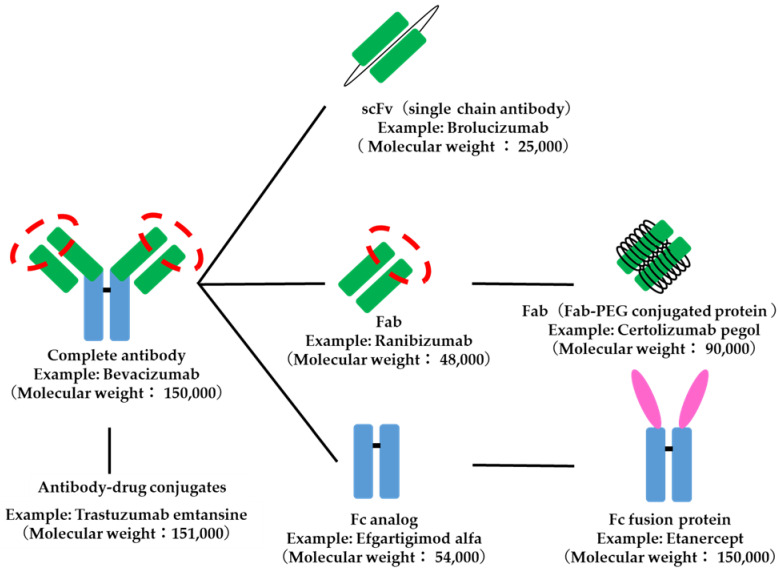
Structure of antibody-related drugs. The red dotted circle indicates the antigen recognition site.

**Figure 7 biomedicines-11-01456-f007:**
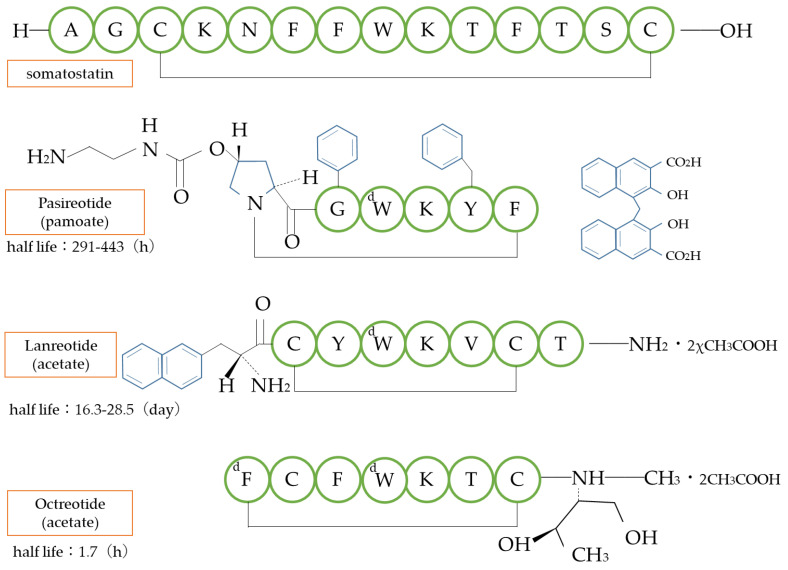
Structure of somatostatin analogs. d: *d*-amino acid type.

**Table 1 biomedicines-11-01456-t001:** Example of classical biopharmaceuticals.

Drug Name	Origin	Main Indications	*t* _1/2_	
Human immunoglobulin	Plasma	No, or hypogammaglobulinemia	21–48	day
Human serum albumin	Plasma	Hypoalbuminemia, hemorrhagic shock	15–20	day
Human alpha 1-proteinase inhibitor	Plasma	Severe alpha 1-antitrypsin deficiency	150.4	h
Human fibrinogen	Plasma	Bleeding tendency in congenital hypofibrinogenemia	3.3–4.2	day
Human antithrombin III	Plasma	Tendency to thrombus formation based on congenital antithrombin III deficiency	65	h
Human haptoglobin	Plasma	Hemoglobinemia, hemoglobinuria	20	h
Human blood coagulation factor	Plasma	Bleeding tendency in patients with blood coagulation factor deficiency	11.6–25.6	h
Human protrobin complex	Plasma	Bleeding tendency during surgery and procedures requiring urgent care	4.2–59.7	h
Human activated protein C	Plasma	Deep vein thrombosis, acute pulmonary thromboembolism	71.5	min
Urinastatin	Urine	Acute exacerbation phase of acute pancreatitis and chronic recurrent pancreatitis	40	min
Urokinase	Urine	Cerebral thrombosis	17–33	min

**Table 2 biomedicines-11-01456-t002:** Serum concentration changes after intravenous administration of Tocilizumab in humans.

Dosage (mg/kg)	C_max_ (μg/mL)	AUC_last_ (μg·h/mL)	*t*_1/2_ (h)	CL_total_ (mL/h/kg)	MRT (h)	Vd,ss (mL/kg)
0.15	2.4 ± 0.6	11 ± 6	17 ± 16	3.8 ± 2.3	25 ± 22	63.4 ± 16.6
0.5	8.5 ± 1.2	285 ± 73	33 ± 4	1.3 ± 0.2	47 ± 5	58.4 ± 7.1
1.0	19.5 ± 2.7	1009 ± 222	49 ± 5	0.8 ± 0.1	69 ± 8	57.3 ± 10.9
2.0	37.6 ± 8.8	2532 ± 569	74 ± 9	0.6 ± 0.2	107 ± 16	65.9 ± 8.3

Quoted from reference [13].

**Table 3 biomedicines-11-01456-t003:** Biopharmaceutical kinetic innovations.

		Example	Note
Control of half life			
	Amino acid modifier	Insulin analog	Insulin lispro	Several amino acids were modified to prolong or shorten the half-life
			Insulin glargine
	PEG modifier	Cytokine	Pegfilgrastim	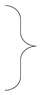	Improved stability against proteolytic enzymes Prolonged half-life due to inhibition of glomerular filtration Improved water solubility as a formulation advantage Reduced antigenicity as a safety benefit

	Fatty acid modifier	GLP-1 analog	Liraglutide

	Albumin fusion protein	Blood coagulation factor	Albutrepenonacog alfa

	Fc fusion protein	Enzyme	Veraglucerase alfa	Proteins that fuse functional proteins such as receptors, peptides, and enzymes with the Fc domains of antibodies
	Fc analog		Efgartigimod alfa	Internal stability
Targeting			
	Fab Analog		Ranibizumab	Targeting to the tissues on a kinetic basis

	Receptor analogue	TNF-alpha antagonist	Etanercept	Fc fusion protein in which the extracellular sequence (domain) of the TNF receptor is bound to the Fc region of human IgG.
Control of half life/Targeting		
	Monoclonal antibody		Adalimumab	A single antibody that binds specifically to disease-related molecules.Takes advantage of the antibody’s unique targeting and long half-life
	Antibody-drug conjugates	Trastuzumab emtansine	Complexes of monoclonal antibodies with specific drugs

**Table 4 biomedicines-11-01456-t004:** Pharmacokinetic characteristics of insulin preparations.

		Drug Name	Daily Dose Number of Times	Expression of Action Pattern (h)	Onset of Action	Maximum Action Expression (h)	Duration of Action(h)
human insulin preparation	fast-acting	insulin	0 3 (Before every meal)	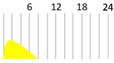	0.5 (h) *	1–3	8 *
	intermediate type	Insulin (protamine preparation 30% mixture)	1-2	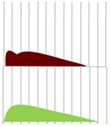	0.5 (h) *	2–8	24 *
		insulin (protamine preparation)	1 (Before breakfast)		1.5 (h) *	4–12	24 *
insulin analog		insulin lispro	3 (Before every meal)	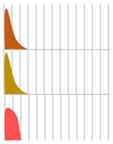	<0.25 (h)	0.5–1.5	3–5
	superquick type	insulin glulisine	3 (Before every meal)		<0.25 (h)	0.5–1.5	3–5
		insulin asparte	3 (Before every meal)		10–20 (m)	1–3	3–5
	mixture type (intermediate type)	insulin asparte (protamine preparation 30% mixture)	1 (Before breakfast)	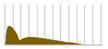	10–20 (m)	1–4	24 *
		insulin asparte (protamine preparation 50% mixture)	1 (Before breakfast)	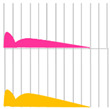	10–20 (m)	1–4	24 *
		insulin asparte (protamine preparation 70% mixture)	1 (Before breakfast)		0.5–1 (h)	1–4	24 *
		insulin glargine	1 (Before breakfast)	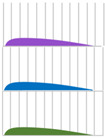	1.11 (h)	–	24 *
	continuation type	insulin detemir	1 (Before dinner)		1.0 (h)*	3–14	24 *
		insulin degludec	1		stationary	–	>42

* approximately.

**Table 5 biomedicines-11-01456-t005:** Examples of pharmacokinetic parameters of monoclonal antibodies.

Drug Name	Indications	Dose		Dosing Interval (Week)	Route *^1^	Parameters *^2^							
						*t* _1/2_		CL_tot_		Vd(mL/kg)		BA(%)	Reference
Adalimumab	rheumatism	40	mg/body	2	sc	12.4	h	18.0	mL/h				[25]
Pembrolizumab	cancer	200	mg/body	3	iv	18.4	day	2.46	mL/day/kg	65.3	mL/kg		[26]
Nivolumab	cancer	2	mg/kg	3	iv	13.3	h	5.04	mL/day/kg	69.7	mL/kg		[27]
Bevacizumab	cancer	5	mg/kg	2<	iv	13.40	day	3.94	mL/day/kg	73.47	mL/kg		[28]
Rituximab	cancer	375	mg/m^2^	1	iv	16.2	h	-	-			-	[29]
Ustekinumab	psoriasis	390	mg/body	8	iv	24.7	day	-	-			-	[30]
Trastuzumab	cancer	4	mg/kg	1	iv	5.9	day	7.4	mL/day/kg	63	mL/kg		[31]
Infliximab	rheumatism	3	mg	2–6	iv	9.5	day	-	-	3	L		[32]
Pertuzumab	cancer	840	mg/body	3	iv	16.8	day	4.25	mL/day/kg	94.1	mL/kg		[33]
Vedolizumab	ulcerative colitis	300	mg/body	1–8	iv	9.46	day	0.258	L/day	3.50	L		[34]
Daratumumab	cancer	16	mg/kg	1–4	iv	17.0	h	3.15	mL/day/kg	72	mL/kg		[35]
Certolizumab pegol	rheumatism	400	mg	1–4	sc	10.7	day	-	-	-	-	-	[36,37]
Cetuximab	rectumcancer	400	mg/m^2^	1	iv	101	h	0.016	L/h/m^2^	2.14	L/m^2^		[38]
Ofatumumab	chronic lymphocytic leukemia	300	mg	1	iv	10	h	199.2	mL/h				[39]
Alemtuzumab	chronic lymphocytic leukemia	3	mg	7	iv	24.06	h	37.7	mL/h/kg				[40]
Mogamulizumab	T-cell leukemia-lymphoma	1	mg/kg	1	iv	422	h						[41]
Panitumumab	cancer	6	mg/kg	2	iv	6.72	day	8.49	mL/day/kg				[42]
Denosumab	osteoporosis	60	mg	6	sc							62	[43]
Tocilizumab	rheumatism	8	mg/kg	4	iv	160	h	0.6	mL/h/kg	137	mL/kg		[44]
Dupilumab	atopic dermatitis	600	mg	2	sc	8.77	day					61–64	[45]
Golimumab	rheumatism	50	mg	4	sc	11.92	day					51	[46]
Ranibizumab	macular degeneration	0.5	mg	1	the vitreous								[47]
Omalizumab	asthma	75–600	mg	2–4	sc	21	day					62–71	[48]
Natalizumab	multiple sclerosis	300	mg	4	iv	365	h	7.28	mL/h	3.51	L		[49]
Atezolizumab	cancer	1200	mg	3	iv	13	day	0.213	L/day	3.82	L		[50]
Durvalumab	cancer	10	mg/kg	2	iv								[51]
Emicizumab	haemophilia	3	mg/kg	1	sc	28.3–29.0	day					80.4–93.1	[52]
Ixekizumab	psoriasis	160	mg	2	sc	11.4–12.2	day					54–90	[53]

* ^1^ Route of administration: iv (intravenous injection) ,sc (Subcutaneous); * ^2^ Quoted from individual interview forms and references.

**Table 6 biomedicines-11-01456-t006:** Structure of antibody–drug conjugates.

Drug Name *^1^	Linker *^2^	Site of Action	Indications	*t*_1/2_(day) *^3^	Reference
Trastuzumab emtansine	MCClinker (thioether)	DM1 (Maytansine Derivatives)	HER2-positive inoperable or recurrent breast cancer	2.39–3.74	[55]
	Emtansine				
Trastuzumab deruxtecan		Camptothecin Derivatives	HER2-positive inoperable or recurrent breast cancer	5.50	[56]
	Deruxtecan			5.77	
Yttrium (^90^Y) ibritumomab thiuxetan	Tiuxetan	Yttrium (^90^Y)	CD20-positive non-Hodgkin’s lymphoma, etc.	34.7–39.3	[57,58,59]
Indium (^111^In) ibritumamob thiuxetan	Tiuxetan	Indium (^111^In)	Ibritsumamobuchiukisetan Identification of accumulation sites	38.6	[59]
Brentuximab vedotin		MMAE (Monomethyl auristatin E)	CD30-positive non-Hodgkin’s lymphoma, etc.	3.75–4.54	[60,61]
	Vedotin				
Gemtuzumab ozogamicin		Ozogamicin (caricamycin)	CD33-positive acute myelogenous leukemia	51–59	[54]

*^1^ Underlined is the antibody part. *^2^ Linker is called alone (e.g., thiuxetan) or together with the site of action (e.g., emtansine). *^3^ Quoted from individual interview forms and references.

**Table 7 biomedicines-11-01456-t007:** Pharmacokinetic characteristics of blood coagulation factors.

Classification (Stem)		Drug	Dosing Interval	Parameter *				
				*t* _1/2_	V_d_	CL_tot_	BA	Reference
				(h)	(mL/kg)	(mL/hr/kg)	(%)	
Factor VIII’ (–octocog)	Plasmaderivative	dried concentrated human blood coagulation factor VIII		15.1				
		Octocog alpha		13.96			45.8–98.4	[62]
		Octocog beta	2–3 times a week	12.8	52.7			
		Rurioctocog alpha		13.00				[63]
		Lonoctocog alpha	2–3 times a week	14.2	56.7	3.00		
		Turoctocog alpha	Every other day/3 times a week	12.61				[64]
	PEG-modified	Lurioctocog alpha Pegol	Twice a week	14.3	50	2.8		
		Turoctogog alpha pegol	1–2 times a week	19.9	37.7	1.4		
		Damoktokog alpha pegol	Every 8–48h	16.3	42.4	1.83		[65]
	Fc fusion protein	Efraloctocog alpha	Every 3–5 days	19.0	49.1	1.95		[66]
Factor IX (–nonacog)	Plasmaderivative	dried concentrated human blood coagulation factor IX		8.2/20.3(α/β)				
		Nonacog alpha		20.2–24.5				
		Nonakog gamma	Twice a week	24.5–27.9				
	PEG–modified	Nonakog beta pegol		83	47			[67]
	Fc fusion protein	Eftrenoacog alpha	Once a week	5.03/82.12	314.8	3.19		[68]
	Albumin fusion protein	Albutrepenonacog alpha	Once a week	104.2	100			[69]

* Quoted from individual interview forms and individual references.

**Table 8 biomedicines-11-01456-t008:** Pharmacokinetic characteristics of interferons.

Drug Name		Dosing Interval	Route *^1^	Parameter *^2^		
				*t*_1/2_(h)	T_max_(h)	BA(%)
Interferon	alpha (natural type)	Once every 1–2 days	sc	9.6	6.0	
	alpha 2b	3 times a week	im	5.2	5.7–6	
	beta	Once a day	iv	-	-	-
	beta-1a	Once a week	im	-	13.0	
	beta 1b	every other day	sc	-	-	-
	beta gamma 1a	Once a day	iv	12.82	8.3	
Peginterferon alpha-2a		Once a week	sc	32.5–42.8	70.9–73.0	84

*^1^ sc: subcutaneous injection. im: intramuscular injection. iv: intravenous injection. *^2^ Quoted from individual interview forms and references.

## Data Availability

Not applicable.

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
