# Peer review of "Pharmacokinetics of Biopharmaceuticals: Their Critical Role in Molecular Design"

_biomedicines, 2023, doi:10.3390/biomedicines11051456_

Round 1

Author Response

R1

Manuscript Number: biomedicines-2323466

Manuscript title: Recent Pharmacokinetic Innovation of Biopharmaceuticals.

The topic taken up by the Authors is very interesting and at the same time needed in the era of the current "quiet" revolution in pharmacology. As the Authors pointed out at the beginning - biopharmaceutics currently account for almost half of all drugs with the highest revenues of pharmaceutical companies. Thanks to technological progress, the number of biopharmaceutics registrations is growing year by year. Nevertheless, despite the growing market of biopharmaceutics, there is relatively little data on their pharmacokinetics. This paper is all the more interesting because it shows how certain "innovations" in the structure of these drugs affect the pharmacokinetics, especially since the pharmacokinetics of biopharmaceutics differs in many ranges from small molecule drugs.

Although the manuscript describe very interesting area, there are many issues that Authors should address in order to improve the accessibility of the entire manuscript to a wider audience:

  1. Major The biggest issue is the “construction” of the manuscript. If the Authors used "pharmacokinetic innovation" in the title of the manuscript, they should collect all these "innovations" (or group them), characterize, and then list the drugs that are characterized by these innovations. Then, if the Authors want to characterize individual drugs (as they do in a significant part of the article), they should pay attention to how a given innovation may affect the “normal” pharmacokinetics of the drug, comparing it to a drug without "this" innovation, if it is of course possible.

It seems to me that such a classification of innovations could be systematized in chapter/section 3. The authors probably intended this in mentioned chapter, although they did not quite do it correctly - this chapter is just “a little messy".

In order to facilitate the "classification of pharmacokinetic innovations", the Authors could first briefly classify biopharmaceutics in the introduction. This will make your work much more readable. So, summing up this remark, the Authors could a) classify biopharmaceutics, then b) present their pharmacokinetics as the Authors do next (with some corrections), c) prepare "pharmacokinetic innovations" and then d) describe the impact of these innovations on the modification of "traditional pharmacokinetics". Such an approach to the topic will very well explain the entire sequence of interrelated elements, which will definitely improve the quality of the manuscript. In addition, the use of the suggested layout of the article will automatically lead to the conclusion presented by the Authors. In its current form, the conclusions are not entirely clear from the content of the article.

 (Reply) Thank you very much for your suggestion.

We will follow your instructions. The title has been changed to Pharmacokinetics of Biopharmaceuticals : their critical role in molecular design. We have also changed the structure of chapter 3, as described below.

After the introduction in Chapter 1, Chapter 2 describes the pharmacokinetics of classical biopharmaceuticals, and Chapter 3 is dedicated to "Quantitative methods for macromolecular pharmaceuticals". From Chapter 4 onwards, the pharmacokinetic characteristics (biostability and targeting) of genetically engineered pharmaceuticals, as typified by antibody drugs, are described. However, as you pointed out, the division of the description between Chapters 2 and 4 is unclear. Therefore, at the beginning of Chapter 2, we have stated that we will first describe the orthodox in vivo pharmacokinetics of classical biologically derived pharmaceuticals.

(Addition in L42-48

The classic biopharamaceuticals is a blood products (plasma fractionated products). Blood coagulation factor VIII and factor IX, which are indicated for hemophilia, have traditionally been derived from fractionated human plasma. Ulinastatin and other drugs derived from human urine are also commercially available. This chapter describes the in vivo kinetics of common biopharmaceuticals, with an emphasis on classical biopharamaceuticals.

I don't think if the “biokinetics” is the right word for this paper (e.g. line 18, 38). In general, biokinetics is very rarely used in the literature describing the pharmacokinetics because this phrase can be used in other fields of science, such as the mechanics of the movement of living organisms or other.

 (Reply)We have amended Biokinetics to Kinetics.

  1. B) Minor – there are a lot of inaccuracies in presented manuscript
  2. The title is good, however is not fully correlated with content of the article – see comment in major objection.

 (Reply) Thank you very much for your suggestion. We have changed the title to Pharmacokinetics of Biopharmaceuticals : their critical role in molecular design, which is closer to the content.

  1. Abstract – the sentence from lines 17-20 (The review…) is very general and there are a few incomprehensible phrases:
  2. a) What it is “biologics”?

 (Reply) We will remove or replace the ward biologics.

  1. b) From this sentence it seems that “biokinetics” is the pharmacokinetics processes + pharmacokinetics? In my opinion it is wrong. The Pharmacokinetics = Absorption + distribution + metabolism + excretion. As I mention in major comment I think that the biokinetics is not a proper word in this manuscript.

 (Reply) We have amended Biokinetics to Kinetics.

3.The introduction section could be rewritten according the major suggestions. After the sentence from lines 34-35 there should be cited literature about this 47% of drugs.

 (Reply)Thank you very much for your suggestion. I have added references and comments.

According to the author's calculations with reference to the literature [3], of the 143 drugs with global sales of $1 billion or more in 2019, biopharmaceuticals accounted for more than 47% of the total. This has been added to the text.

(Addition in L35-37)The authors calculated, based on a paper [1], that of the 143 drugs with global sales of more than USD 1 billion in 2019,Sales of biopharmaceuticals accounted for more than 47% of the total.

  1. The second section: a) Absorption subsection – In first sentence (lines 40-41) the Authors claim that the absorption of biopharmaceutics is limited by the large molecular weight, while in the next sentence the Authors claim that the absorption is limited because of degradation of biopharmaceutics by the digestive enzymes.

  • In line 46 (and also 66, 88, 323) there is lack of the unit of molecular weight/mass (e.g. Da or u).

 (Reply) In this manuscript, we did not use molecular mass (Da) but molecular weight. Because molecular weight is a relative value and does not have any unit, so we did not add Da after the number that show molecular weight.

[Surprising differences between molecular weight and mass units

https://www.sbj.or.jp/wp-content/uploads/file/sbj/9108/9108_yomoyama.pdf]

  • In line 54 there should be cited literature in the end of the sentence.

 (Reply) We checked the BA of the individual drugs listed in chapter 4 and late found that the BA was generally 40-100%. However, there was no description of BA in the individual tables after Chapter 4. Therefore, we have added BA for drugs other than those administered intravenously as much as possible. Thank you very much for your suggestion.

  • There is lack of summarized of absorption of biopharmaceutics – so which route is the main route of administration of these drugs (the route which is results from the specificity of absorption of these drugs, which in turn results from their physicochemical properties)? Such a summary shows the legitimacy of attempting to modify these drugs in order to improve their absorption (and thus introduce innovative modifications in structure).

 (Reply) Thank you very much for your suggestion. We have entered the route of administration in all tables where possible, and for drugs administered by routes other than intravenous, we have entered the bioavailability. In addition, the figure at the beginning of Chapter 4 has been replaced by a table3 listing the structural characteristics of biopharmaceuticals.

  1. b) Distribution subsection – in line 62 there should be cited literature in the end of the sentence.

 (Reply) Thank you for your comments. Many macromolecular drugs have low membrane permeability and are distributed almost entirely in plasma when administered intravenously, which is why the volume of distribution is often almost equal to the plasma volume, as shown in the respective discussion from Chapter 4. We have therefore added this to the tables in Chapter 4 onwards, adding the volume of distribution as far as possible and directing the reader's attention to the reference papers shown.

(Addition in L75-76)In general, drugs derived from high-molecular-weight biomolecules have difficulty permeating biological membranes and have low tissue distribution. Therefore, when ad-ministered intravenously, they tend to remain in the blood, and the volume of distribution is often about 60 mL/kg (body weight), equal to the plasma volume, shown in the individual sections following Chapter 4.

  1. c) Excretion subsection – line 84 – which pharmaceuticals? Please give some examples.

 (Reply) Thank you for your comments. Many medicinal products derived from biological components are excreted in the urine after being reduced in molecular weight by metabolism; the Excretion subsection has been replaced by the following sentences.

 (Corrected in L95-106).

Many pharmaceuticals derived from biocomponents are excreted in the urine after being metabolised into lower molecular weight compounds. Most of the examples given from Chapter 4 onwards are not excreted in the urine in their unchanged form. As with low-molecular-weight drugs, the ease of glomerular filtration of macromolecular drugs is related to their molecular size and charge, and the rate of filtration decreases markedly when the molecular weight exceeds 30,000. In addition, positively charged drugs are more easily filtered [9,10]. Relatively low-molecular-weight proteins filtered from the glomerulus are reabsorbed by endocytosis in the proximal tubules and broken down into amino acids in the tubular cells. Therefore, the unchanged form is rarely excreted in the urine, and even if it is, only a few per cent of the dose is excreted. However, the urinary excretion rate of preparations purified from human urine is high: 24% of the dose of urinastatin (Table 1) is excreted in the urine.

  • in line 88-89 there should be cited literature in the end of the sentence about influence of positive charge on excretion.

 (Reply) Thank you very much for your comments. We have listed two papers, including one on urinastatin, for which we were involved in drug development.

(Addition in reference)

Eur J Nucl Med. 1998 25, 201-12. Reducing the renal uptake of radiolabeled antibody fragments and peptides for diagnosis and therapy: present status, future prospects and limitations. T M Behr 1, D M Goldenberg, W Becker

  • Authors should define the “urinastatin”, or describe which groups of biopharmaceutics (if the Authors prepare some classification) are excreted via urine in changed or unchanged form.

 (Reply) We have stated that urinastatin is excreted in the urine as an unchanged drug.

This drug is a positively charged glycoprotein with a molecular weight of 67,000, originally purified from urine, and is known as a drug for acute and chronic recurrent pancreatitis because it inhibits various pancreatic enzymes as well as trypsin. It is known to be excreted in urine in an unchanged form. However, as it is marketed only in Japan, it is not well known globally. The item 'Excretion' has been amended as above to include this point.

  1. d) Pharmacokinetics subsection – maybe better will be “Selected pharmacokinetic parameters of biopharmaceutics”? The description of table 1 is inadequate – this table is cited for half-life parameter.

 (Reply)Thank you very much for your comments. As mentioned above, the structure of this paper is as follows: after the introduction (marketability of biopharmaceuticals) in chapter 1, the pharmacokinetics of classical biopharmaceuticals is described in chapter 2, and from chapter 4 onwards, the pharmacokinetic characteristics (biostability and targeting) of genetically engineered pharmaceuticals, as typified by antibody drugs, are described. targeting) of genetically engineered medicinal products, such as antibody medicinal products.

However, the positioning of Chapters 2 and 4 was unclear, resulting in a misleading structure for the reader. To clarify the position of Chapter 2, the aforementioned sentence was inserted at the beginning of Chapter 2. Furthermore, this session and the subsequent Pharmacokinetics session have been combined as drug ‘Elimination'.

  1. e) in the second paragraph of this subsection Authors mixed the descrition of the excretion with parameters. This is due the fact that the Authors first describe the pharmacokinetic processes, and in this subsection the probably do not know what add. Maybe Authors will combine this subsection with excretion? It will be more clearly. Additionally, if the figure 1 is from other studies there should be citation about it.

 (Reply) As mentioned above, the session on excretion and Pharmacokinetics were combined as 'elimination' of drugs.

  1. f) In reviewer opinion the Drug quantification section is not suitable for whole manuscript – if the Authors are focused on the pharmacokinetics they should not mixed it with analytical tools.

(Reply) Thank you very much for your comments. The method of quantification of pharmaceuticals is closely related to their pharmacokinetics. In addition, the quantification methods for biological products in vivo are different from those for small molecule pharmaceuticals and are quite specific. However, as you pointed out, it seems a little inappropriate to mention quantification methods in this subsection. We have therefore decided to make it an independent section.

  1. The third section:
  2. a) If the Authors will use my comments about the structure of the manuscript, than this section will have to be rewrite. First if the Authors should precise what is the pharmacokinetic innovation, than how this innovation influence on pharmacokinetic. The Authors can do it during description of each biopharmaceutics class e.g. First Authors describe the monoclonal antibodies – the structure, and the influence of the structure or changed in structure on PK etc. Next the insulins the same – structure, changes in the structures and influence of these changes on pharmacokinetics. Next the blood coagulation factors etc.

 (Reply) Thank you very much for your suggestion. Firstly, we have changed the title of the article according to your instructions. We have also added a figure on the structure and characteristics of insulin.

You indicated that we should replace the sections on antibody-related medicinal products and insulin. However, antibody-related medicinal products are not limited to monoclonal antibodies alone, but also include Fc and Fab analogues and antibody-drug conjugates. Furthermore, there are PEGylated pharmaceuticals that are expected to have a longer half-life. For this reason, insulin, which is relatively easy to describe independently, is described at the beginning of this article. We hope you will understand.

  1. b) If the figures are from other manuscript it should be citation.

 (Reply) This paper is an English translation of a Japanese paper written by me, with updated information and additions. Permission for this has been obtained from both publishers. I have added this to the acknowledgements of the earlier paper. In addition, all figures and tables except for Tocilizumab were prepared by me and are copyrighted by me. I have added this copyright notice in this new version.

  1. c) Authors should use the same font types and size in the figures.

(Reply) We have followed your instructions and standardised the font type and size used in the figures.

  1. d) In the References section there are also small inaccuracies like to many spaces (e.g. line 402, 415 and other)

 (Reply) We have followed your instructions and standardised the font type and size used in the figures.

  1. Other minor comments
  2. a) All of the tables should be formatted according one template – in each table there are mistakes in fonts etc.

 (Reply) We have followed your instructions and standardised the font type used in the figures.

R2

The submitted review describes some aspects of biopharmaceutical preparations. The review is neither comprehensive nor systematic, instead it presents information from various branches without going into details. The figures of this review are of high quality, however it is not entirely clear if they have been reproduced from other works (which is typical for review works) or were created for the purpose of the current manuscript – this must be addressed.

 (Addition in L399-400) This review is based on the Japanese edition of Pharmacokinetics, a textbook for pharmacists published by Kyoto-Hirokawa Publishing Inc., and has been updated to include the latest findings. We thank the editors of Biomedicines and Kyoto-Hirokawa Publishing Inc. for their permission and understanding. The authors retain authorship of all original figures and tables cited in this paper.

Line 13, what do you mean by “basic structure”?

 (Reply) We have deleted “basic” because its meaning is ambiguous.

Lines 34-35, the references are needed here. Also, the Authors refer to 2019, it would be reasonable to update those values

 (Reply)Thank you very much for your suggestion. I have added references and comments.

According to the author's calculations with reference to the literature [3], of the 143 drugs with global sales of $1 billion or more in 2019, biopharmaceuticals accounted for more than 47% of the total. This has been added to the text.

(Added in L35-37)The authors calculated based on the paper [1], that of the 143 drugs with global sales of $1 billion or more in 2019, biopharmaceuticals accounted for more than 47% of the total.

Section 2, what about the solubility?

 (Reply) This review describes the pharmacokinetics of biopharmaceuticals and 'solubility' was not mentioned because it is a physical or formulation factor.

Figure 1, does this figure originate from other work or was it created for the purpose of this review?

(Reply) Cited references are included.

Tocilizumab interview form.

Line 118, it should be 125I

 (Reply) We have corrected it according to your suggestion.

The number of references is scarce. I.e., page 5, despite a lot of information presented on this page there are not any references.

 (Reply) Following your suggestion, we have increased the number of references. In particular, we have increased the number of examples of antibody drugs in Table 5 and added the corresponding references. The reference paper on the description on page 5 has been added as a reference paper in each of the chapters from Chapter 4 onwards.

Despite the title, suggesting recent innovations, there a lot of historical information presented in multiple places, i.e. the history of insulin. This should be revised.

 (Reply) The title has been changed in accordance with your suggestion. Duplicated sections of Insurin's history have been removed.

The style of tables must be unified, i.e. Table 4 and Table 5 look very different.

 (Reply) We have corrected it according to your suggestion.

R3

The manuscript entitled “Recent Pharmacokinetic Innovation of Biopharmaceuticals” reviews the pharmacokinetic aspects of some drugs derived from biological components.

Although the topic is interesting, the manuscript does not provide the reader with an in-depth analysis of the factors involved in determining the pharmacokinetic profile of new developed biopharmaceuticals.

Reviewer 2 Report

The submitted review describes some aspects of biopharmaceutical preparations. The review is neither comprehensive nor systematic, instead it presents information from various branches without going into details. The figures of this review are of high quality, however it is not entirely clear if they have been reproduced from other works (which is typical for review works) or were created for the purpose of the current manuscript – this must be addressed.

Line 13, what do you mean by “basic structure”?

Lines 34-35, the references is needed here. Also, the Authors refer to 2019, it would be reasonable to update those values

Section 2, what about the solubility?

Figure 1, does this figure originate from other work or was it created for the purpose of this review?

Line 118, it should be 125I

The number of references is scarce. I.e., page 5, despite a lot of information presented on this page there are not any references.

Despite the title, suggesting recent innovations, there a lot of historical information presented in multiple places, i.e. the history of insulin. This should be revised.

The style of tables must be unified, i.e. Table 4 and Table 5 look very different.

Author Response

(The authors gave the same response as above.)

Reviewer 3 Report

The manuscript entitled “Recent Pharmacokinetic Innovation of Biopharmaceuticals” reviews the pharmacokinetic aspects of some drugs derived from biological components.

Although the topic is interesting, the manuscript does not provide the reader with an in-depth analysis of the factors involved in determining the pharmacokinetic profile of new developed biopharmaceuticals.

Author Response

R1

Manuscript Number: biomedicines-2323466

Manuscript title: Recent Pharmacokinetic Innovation of Biopharmaceuticals.

The topic taken up by the Authors is very interesting and at the same time needed in the era of the current "quiet" revolution in pharmacology. As the Authors pointed out at the beginning - biopharmaceutics currently account for almost half of all drugs with the highest revenues of pharmaceutical companies. Thanks to technological progress, the number of biopharmaceutics registrations is growing year by year. Nevertheless, despite the growing market of biopharmaceutics, there is relatively little data on their pharmacokinetics. This paper is all the more interesting because it shows how certain "innovations" in the structure of these drugs affect the pharmacokinetics, especially since the pharmacokinetics of biopharmaceutics differs in many ranges from small molecule drugs.

Although the manuscript describe very interesting area, there are many issues that Authors should address in order to improve the accessibility of the entire manuscript to a wider audience:

  1. Major The biggest issue is the “construction” of the manuscript. If the Authors used "pharmacokinetic innovation" in the title of the manuscript, they should collect all these "innovations" (or group them), characterize, and then list the drugs that are characterized by these innovations. Then, if the Authors want to characterize individual drugs (as they do in a significant part of the article), they should pay attention to how a given innovation may affect the “normal” pharmacokinetics of the drug, comparing it to a drug without "this" innovation, if it is of course possible.

It seems to me that such a classification of innovations could be systematized in chapter/section 3. The authors probably intended this in mentioned chapter, although they did not quite do it correctly - this chapter is just “a little messy".

In order to facilitate the "classification of pharmacokinetic innovations", the Authors could first briefly classify biopharmaceutics in the introduction. This will make your work much more readable. So, summing up this remark, the Authors could a) classify biopharmaceutics, then b) present their pharmacokinetics as the Authors do next (with some corrections), c) prepare "pharmacokinetic innovations" and then d) describe the impact of these innovations on the modification of "traditional pharmacokinetics". Such an approach to the topic will very well explain the entire sequence of interrelated elements, which will definitely improve the quality of the manuscript. In addition, the use of the suggested layout of the article will automatically lead to the conclusion presented by the Authors. In its current form, the conclusions are not entirely clear from the content of the article.

 (Reply) Thank you very much for your suggestion.

We will follow your instructions. The title has been changed to Pharmacokinetics of Biopharmaceuticals : their critical role in molecular design. We have also changed the structure of chapter 3, as described below.

After the introduction in Chapter 1, Chapter 2 describes the pharmacokinetics of classical biopharmaceuticals, and Chapter 3 is dedicated to "Quantitative methods for macromolecular pharmaceuticals". From Chapter 4 onwards, the pharmacokinetic characteristics (biostability and targeting) of genetically engineered pharmaceuticals, as typified by antibody drugs, are described. However, as you pointed out, the division of the description between Chapters 2 and 4 is unclear. Therefore, at the beginning of Chapter 2, we have stated that we will first describe the orthodox in vivo pharmacokinetics of classical biologically derived pharmaceuticals.

(Addition in L42-48

The classic biopharamaceuticals is a blood products (plasma fractionated products). Blood coagulation factor VIII and factor IX, which are indicated for hemophilia, have traditionally been derived from fractionated human plasma. Ulinastatin and other drugs derived from human urine are also commercially available. This chapter describes the in vivo kinetics of common biopharmaceuticals, with an emphasis on classical biopharamaceuticals.

I don't think if the “biokinetics” is the right word for this paper (e.g. line 18, 38). In general, biokinetics is very rarely used in the literature describing the pharmacokinetics because this phrase can be used in other fields of science, such as the mechanics of the movement of living organisms or other.

 (Reply)We have amended Biokinetics to Kinetics.

  1. B) Minor – there are a lot of inaccuracies in presented manuscript
  2. The title is good, however is not fully correlated with content of the article – see comment in major objection.

 (Reply) Thank you very much for your suggestion. We have changed the title to Pharmacokinetics of Biopharmaceuticals : their critical role in molecular design, which is closer to the content.

  1. Abstract – the sentence from lines 17-20 (The review…) is very general and there are a few incomprehensible phrases:
  2. a) What it is “biologics”?

 (Reply) We will remove or replace the ward biologics.

  1. b) From this sentence it seems that “biokinetics” is the pharmacokinetics processes + pharmacokinetics? In my opinion it is wrong. The Pharmacokinetics = Absorption + distribution + metabolism + excretion. As I mention in major comment I think that the biokinetics is not a proper word in this manuscript.

 (Reply) We have amended Biokinetics to Kinetics.

3.The introduction section could be rewritten according the major suggestions. After the sentence from lines 34-35 there should be cited literature about this 47% of drugs.

 (Reply)Thank you very much for your suggestion. I have added references and comments.

According to the author's calculations with reference to the literature [3], of the 143 drugs with global sales of $1 billion or more in 2019, biopharmaceuticals accounted for more than 47% of the total. This has been added to the text.

(Addition in L35-37)The authors calculated, based on a paper [1], that of the 143 drugs with global sales of more than USD 1 billion in 2019,Sales of biopharmaceuticals accounted for more than 47% of the total.

  1. The second section: a) Absorption subsection – In first sentence (lines 40-41) the Authors claim that the absorption of biopharmaceutics is limited by the large molecular weight, while in the next sentence the Authors claim that the absorption is limited because of degradation of biopharmaceutics by the digestive enzymes.

  • In line 46 (and also 66, 88, 323) there is lack of the unit of molecular weight/mass (e.g. Da or u).

 (Reply) In this manuscript, we did not use molecular mass (Da) but molecular weight. Because molecular weight is a relative value and does not have any unit, so we did not add Da after the number that show molecular weight.

[Surprising differences between molecular weight and mass units

https://www.sbj.or.jp/wp-content/uploads/file/sbj/9108/9108_yomoyama.pdf]

  • In line 54 there should be cited literature in the end of the sentence.

 (Reply) We checked the BA of the individual drugs listed in chapter 4 and late found that the BA was generally 40-100%. However, there was no description of BA in the individual tables after Chapter 4. Therefore, we have added BA for drugs other than those administered intravenously as much as possible. Thank you very much for your suggestion.

  • There is lack of summarized of absorption of biopharmaceutics – so which route is the main route of administration of these drugs (the route which is results from the specificity of absorption of these drugs, which in turn results from their physicochemical properties)? Such a summary shows the legitimacy of attempting to modify these drugs in order to improve their absorption (and thus introduce innovative modifications in structure).

 (Reply) Thank you very much for your suggestion. We have entered the route of administration in all tables where possible, and for drugs administered by routes other than intravenous, we have entered the bioavailability. In addition, the figure at the beginning of Chapter 4 has been replaced by a table3 listing the structural characteristics of biopharmaceuticals.

  1. b) Distribution subsection – in line 62 there should be cited literature in the end of the sentence.

 (Reply) Thank you for your comments. Many macromolecular drugs have low membrane permeability and are distributed almost entirely in plasma when administered intravenously, which is why the volume of distribution is often almost equal to the plasma volume, as shown in the respective discussion from Chapter 4. We have therefore added this to the tables in Chapter 4 onwards, adding the volume of distribution as far as possible and directing the reader's attention to the reference papers shown.

(Addition in L75-76)In general, drugs derived from high-molecular-weight biomolecules have difficulty permeating biological membranes and have low tissue distribution. Therefore, when ad-ministered intravenously, they tend to remain in the blood, and the volume of distribution is often about 60 mL/kg (body weight), equal to the plasma volume, shown in the individual sections following Chapter 4.

  1. c) Excretion subsection – line 84 – which pharmaceuticals? Please give some examples.

 (Reply) Thank you for your comments. Many medicinal products derived from biological components are excreted in the urine after being reduced in molecular weight by metabolism; the Excretion subsection has been replaced by the following sentences.

 (Corrected in L95-106).

Many pharmaceuticals derived from biocomponents are excreted in the urine after being metabolised into lower molecular weight compounds. Most of the examples given from Chapter 4 onwards are not excreted in the urine in their unchanged form. As with low-molecular-weight drugs, the ease of glomerular filtration of macromolecular drugs is related to their molecular size and charge, and the rate of filtration decreases markedly when the molecular weight exceeds 30,000. In addition, positively charged drugs are more easily filtered [9,10]. Relatively low-molecular-weight proteins filtered from the glomerulus are reabsorbed by endocytosis in the proximal tubules and broken down into amino acids in the tubular cells. Therefore, the unchanged form is rarely excreted in the urine, and even if it is, only a few per cent of the dose is excreted. However, the urinary excretion rate of preparations purified from human urine is high: 24% of the dose of urinastatin (Table 1) is excreted in the urine.

  • in line 88-89 there should be cited literature in the end of the sentence about influence of positive charge on excretion.

 (Reply) Thank you very much for your comments. We have listed two papers, including one on urinastatin, for which we were involved in drug development.

(Addition in reference)

Eur J Nucl Med. 1998 25, 201-12. Reducing the renal uptake of radiolabeled antibody fragments and peptides for diagnosis and therapy: present status, future prospects and limitations. T M Behr 1, D M Goldenberg, W Becker

  • Authors should define the “urinastatin”, or describe which groups of biopharmaceutics (if the Authors prepare some classification) are excreted via urine in changed or unchanged form.

 (Reply) We have stated that urinastatin is excreted in the urine as an unchanged drug.

This drug is a positively charged glycoprotein with a molecular weight of 67,000, originally purified from urine, and is known as a drug for acute and chronic recurrent pancreatitis because it inhibits various pancreatic enzymes as well as trypsin. It is known to be excreted in urine in an unchanged form. However, as it is marketed only in Japan, it is not well known globally. The item 'Excretion' has been amended as above to include this point.

  1. d) Pharmacokinetics subsection – maybe better will be “Selected pharmacokinetic parameters of biopharmaceutics”? The description of table 1 is inadequate – this table is cited for half-life parameter.

 (Reply)Thank you very much for your comments. As mentioned above, the structure of this paper is as follows: after the introduction (marketability of biopharmaceuticals) in chapter 1, the pharmacokinetics of classical biopharmaceuticals is described in chapter 2, and from chapter 4 onwards, the pharmacokinetic characteristics (biostability and targeting) of genetically engineered pharmaceuticals, as typified by antibody drugs, are described. targeting) of genetically engineered medicinal products, such as antibody medicinal products.

However, the positioning of Chapters 2 and 4 was unclear, resulting in a misleading structure for the reader. To clarify the position of Chapter 2, the aforementioned sentence was inserted at the beginning of Chapter 2. Furthermore, this session and the subsequent Pharmacokinetics session have been combined as drug ‘Elimination'.

  1. e) in the second paragraph of this subsection Authors mixed the descrition of the excretion with parameters. This is due the fact that the Authors first describe the pharmacokinetic processes, and in this subsection the probably do not know what add. Maybe Authors will combine this subsection with excretion? It will be more clearly. Additionally, if the figure 1 is from other studies there should be citation about it.

 (Reply) As mentioned above, the session on excretion and Pharmacokinetics were combined as 'elimination' of drugs.

  1. f) In reviewer opinion the Drug quantification section is not suitable for whole manuscript – if the Authors are focused on the pharmacokinetics they should not mixed it with analytical tools.

(Reply) Thank you very much for your comments. The method of quantification of pharmaceuticals is closely related to their pharmacokinetics. In addition, the quantification methods for biological products in vivo are different from those for small molecule pharmaceuticals and are quite specific. However, as you pointed out, it seems a little inappropriate to mention quantification methods in this subsection. We have therefore decided to make it an independent section.

  1. The third section:
  2. a) If the Authors will use my comments about the structure of the manuscript, than this section will have to be rewrite. First if the Authors should precise what is the pharmacokinetic innovation, than how this innovation influence on pharmacokinetic. The Authors can do it during description of each biopharmaceutics class e.g. First Authors describe the monoclonal antibodies – the structure, and the influence of the structure or changed in structure on PK etc. Next the insulins the same – structure, changes in the structures and influence of these changes on pharmacokinetics. Next the blood coagulation factors etc.

 (Reply) Thank you very much for your suggestion. Firstly, we have changed the title of the article according to your instructions. We have also added a figure on the structure and characteristics of insulin.

You indicated that we should replace the sections on antibody-related medicinal products and insulin. However, antibody-related medicinal products are not limited to monoclonal antibodies alone, but also include Fc and Fab analogues and antibody-drug conjugates. Furthermore, there are PEGylated pharmaceuticals that are expected to have a longer half-life. For this reason, insulin, which is relatively easy to describe independently, is described at the beginning of this article. We hope you will understand.

  1. b) If the figures are from other manuscript it should be citation.

 (Reply) This paper is an English translation of a Japanese paper written by me, with updated information and additions. Permission for this has been obtained from both publishers. I have added this to the acknowledgements of the earlier paper. In addition, all figures and tables except for Tocilizumab were prepared by me and are copyrighted by me. I have added this copyright notice in this new version.

  1. c) Authors should use the same font types and size in the figures.

(Reply) We have followed your instructions and standardised the font type and size used in the figures.

  1. d) In the References section there are also small inaccuracies like to many spaces (e.g. line 402, 415 and other)

 (Reply) We have followed your instructions and standardised the font type and size used in the figures.

  1. Other minor comments
  2. a) All of the tables should be formatted according one template – in each table there are mistakes in fonts etc.

 (Reply) We have followed your instructions and standardised the font type used in the figures.

R2

The submitted review describes some aspects of biopharmaceutical preparations. The review is neither comprehensive nor systematic, instead it presents information from various branches without going into details. The figures of this review are of high quality, however it is not entirely clear if they have been reproduced from other works (which is typical for review works) or were created for the purpose of the current manuscript – this must be addressed.

 (Addition in L399-400) This review is based on the Japanese edition of Pharmacokinetics, a textbook for pharmacists published by Kyoto-Hirokawa Publishing Inc., and has been updated to include the latest findings. We thank the editors of Biomedicines and Kyoto-Hirokawa Publishing Inc. for their permission and understanding. The authors retain authorship of all original figures and tables cited in this paper.

Line 13, what do you mean by “basic structure”?

 (Reply) We have deleted “basic” because its meaning is ambiguous.

Lines 34-35, the references are needed here. Also, the Authors refer to 2019, it would be reasonable to update those values

 (Reply)Thank you very much for your suggestion. I have added references and comments.

According to the author's calculations with reference to the literature [3], of the 143 drugs with global sales of $1 billion or more in 2019, biopharmaceuticals accounted for more than 47% of the total. This has been added to the text.

(Added in L35-37)The authors calculated based on the paper [1], that of the 143 drugs with global sales of $1 billion or more in 2019, biopharmaceuticals accounted for more than 47% of the total.

Section 2, what about the solubility?

 (Reply) This review describes the pharmacokinetics of biopharmaceuticals and 'solubility' was not mentioned because it is a physical or formulation factor.

Figure 1, does this figure originate from other work or was it created for the purpose of this review?

(Reply) Cited references are included.

Tocilizumab interview form.

Line 118, it should be 125I

 (Reply) We have corrected it according to your suggestion.

The number of references is scarce. I.e., page 5, despite a lot of information presented on this page there are not any references.

 (Reply) Following your suggestion, we have increased the number of references. In particular, we have increased the number of examples of antibody drugs in Table 5 and added the corresponding references. The reference paper on the description on page 5 has been added as a reference paper in each of the chapters from Chapter 4 onwards.

Despite the title, suggesting recent innovations, there a lot of historical information presented in multiple places, i.e. the history of insulin. This should be revised.

 (Reply) The title has been changed in accordance with your suggestion. Duplicated sections of Insurin's history have been removed.

The style of tables must be unified, i.e. Table 4 and Table 5 look very different.

 (Reply) We have corrected it according to your suggestion.

R3

The manuscript entitled “Recent Pharmacokinetic Innovation of Biopharmaceuticals” reviews the pharmacokinetic aspects of some drugs derived from biological components.

Although the topic is interesting, the manuscript does not provide the reader with an in-depth analysis of the factors involved in determining the pharmacokinetic profile of new developed biopharmaceuticals.

(reply) 

We have revised it extensively and ask you to review it once more.

Round 2

Reviewer 1 Report

Thank you for considering my comments. I hope this has improved the quality of the article.

Reviewer 2 Report

The Authors have significantly improved their manuscript. This version is acceptable.

Reviewer 3 Report

The authors revised the manuscript properly.